# Three-Dimensional-Printed Fabrication of POFs Using Different Filaments and Their Characterization for Sensing Applications

**DOI:** 10.3390/polym15030640

**Published:** 2023-01-26

**Authors:** Robertson Pires-Junior, Leandro Macedo, Anselmo Frizera, Maria José Pontes, Arnaldo Leal-Junior

**Affiliations:** Graduate Program in Electrical Engineering, Federal University of Espírito Santo, Vitória 29075-910, ES, Brazil

**Keywords:** polymer optical fiber, 3D printing, sensors

## Abstract

This paper presents the development and sensor applications of 3D-printed polymer optical fibers (POFs) using commercially available filaments. The well-known intensity variation sensor was developed using this fiber for temperature and curvature sensing, where the results indicate a linear response in the curvature analysis, with a coefficient of determination (R${}^{2}$) of 0.97 and sensitivity of 4.407 × 10${}^{-4}$ mW/${}^{\circ }$, whereas the temperature response was fitted to an R${}^{2}$ of 0.956 with a sensitivity of 5.718 × 10${}^{-3}$ mW/${}^{\circ }$C. Then, the POF was used in the development of a modal interferometer by splicing the POF in-between two single-mode fibers (SMFs), which result in a single-mode-multimode-single-mode (SMS) configuration. The such interferometer was tested for temperature and axial strain responses, where the temperature response presented a linear trend R${}^{2}$ of around 0.98 with a sensitivity of −78.8 pm/${}^{\circ }$C. The negative value of the sensitivity is related to the negative thermo-optic coefficient commonly obtained in POFs. Furthermore, the strain response of the SMS interferometer showed a high sensitivity (9.5 pm/μϵ) with a quadratic behavior in which the R${}^{2}$ of around 0.99 was obtained. Therefore, the proposed approach is a low-cost, environmentally friendly and straightforward method for the production of highly sensitive optical fiber sensors.

## 1. Introduction

The first application of a polymer optical fiber (POF), developed in the sixties, was focused on data transmission [1]. Despite POFs’ high attenuation compared to silica fibers, they are used to transmit data over short distances [2]. Furthermore, with the advance in technologies and new materials, recent works accomplish that POFs have different sensors applications for mechanical and chemical parameters [3,4]. As compared to silica optical fibers, POF has higher strain limits, making it attractive for sensor applications [3]. In addition, POF’s properties can be tailored according to the application, as different raw polymers with different characteristics can be used to produce them [5].

The different manufacturing techniques for POFs depend on the process, where the techniques can be categorized as continuous or discontinuous [6]. In the first technique, all processes are simultaneous, which enables the production of large quantities of filament. The second technique involves, at least, two steps, limiting the length of produced filament compared to continuous techniques [7]. A few examples of continuous processes are continuous extrusion [1], photochemical polymerization [7], co-extrusion [8], dry spinning [9], and melt spinning [1]. In terms of discontinuous processes, some examples include preform production [4], heat-drawing process [3], and batch extrusion [10].

Innovative technologies to enhance POFs material features involve combining different material characteristics, resulting in a multimaterial POF for the enhancement of their mechanical and optical properties, e.g., the heat-resistant POF in [11]. Furthermore, it is possible to enhance the sensitivity of the POF as a function of different quantities by combining different materials. Such materials with different Young’s modulus result in multimaterial optical fibers, which can also be made relatively insensitive to undesired parameters such as humidity and temperature [12].

Structures with complex geometries can be fabricated from a variety of materials (such as biomaterials, ceramics, composites, polymers, metal alloys) using 3D printing, an additive manufacturing (AM) process [13]. The simplicity, relatively low cost, less waste, freedom of design, and automation of this technology have made it widely explored [14]. Rapid prototyping, printing large structures, and printing complex structures at high resolution all contributed to the development of some AM techniques, which enable its application in some fields, such as biomedical, mechanical, civil construction, and microelectronic device manufacturing [14].

Fused deposition modeling (FDM) is one of the main fabrication methods of AM [6]. In this process, thermoplastic polymers are heated to a semi-liquid state and then extruded onto a platform or on top of previously extruded layers [14]. A key property of polymers is their thermoplasticity, which allows extruded polymers to fuse together and then solidify at room temperature [15]. The mechanical properties of the printed part are strongly affected by some parameters such as: layer thickness, width and orientation of filaments and air gap [16]. While FDM has some advantages, including low cost, high speed, and simplicity, it also has some limitations, including poor mechanical properties, poor surface quality, and layer-by-layer appearance [17].

Other AM methods include powder bed fusion, inkjet printing and contour crafting, stereolithography (SLA), and direct energy deposition [14]. There are advantages and limitations to each of these methods, and they need to be taken into consideration with other parameters such as the size of the part, the surface quality required, and the time it will take to fabricate the part [18].

The progress in the AM methods, especially in 3D printing resulted in many advances in polymer processing, where different printing parameters were thoroughly investigated for conventional materials in 3D printing, e.g., PLA and PETG [19]. The investigation of such parameters not only leads to the possibility of optimization in mechanical properties of 3D printing devices [20], but also in novel shape memory devices [21] and hybrid composites [22]. Such developments lead to the possibility of 4D printing of polymer-based devices [23] for shape memory performance in mechanical and thermal parameters [24]. As another possibility of the AM method in transparent materials, the development of waveguides was explored not only using plastics [25], but also in glass structures [26]. In this scenario, the development of optical devices using polymers [27] and, especially, POFs [28] result in an additional possibility in optical fiber technology. These POFs are generally used in the development of sensors, where the integration of AM methods results in multiparameter sensing [29] as well as the performance enhancement in temperature [30] and force [31] sensors. The practical applications of such fibers also include the development of smart textiles [32], where such textiles can be used for angle measurement [33] and activity monitoring with smart clothing [34].

This work is focused on the fabrication and sensor applications of a POF using the AM process. Mechanical and optical characterization methods are employed for curvature and temperature sensing applications. Additionally, the fabricated POFs are used in the development of a modal interferometer using the 3D-printed POF in-between single-mode fibers for strain and temperature assessment. The progress in the 3D-printing fabrication of POFs involves the use of different optical materials and methods for extrusion as summarized in [28]. In addition, the application of multimaterial approaches were also investigated using temperature-sensitive resins [35] and even 3D-printing approaches [36]. However, the application of such 3D-printed devices in interferometric approaches for sensing was not investigated in the literature yet. Thus, the main contribution of this work is the low cost fabrication of 3D-printed fibers directly extruded from the 3D printer (instead of preforms as discussed in [28]) in conjunction with the sensors applications using not only the intensity variation principle, but also using modal interferometers. The novelty of this approach is not only the customization of the optical fiber fabrication, especially on the cladding fabrication, but also in the novel approach for interferometer development using 3D-printed POFs. The proposed sensor device can be used in different applications ranging from biosensors [37], structural monitoring [38] and yarn-integrated devices [39].

## 2. Fabrication Method

POF’s fabrication process begins with the selection of the materials for its core and cladding. From the 3D printing filaments analyzed, the main characteristic of the fabrication process is transparency. The first analysis is the refractive index measurement of four different filaments: HDglass (a commercial PET-G filament with additives for higher transparency from FormFutura), PET-G (from 3D-Fila, Brazil), PLA (from 3D-Fila, Brazil) and Tritan (a copolyester commercial filament from 3D-Fila, Brazil). The performance of a second test is necessary to measure their transmittance. These filaments were chosen due to their wide commercial availability and the differences in their optical and mechanical properties as well as the differences in their printing parameters such as the nozzle and bed temperatures. The HDglass is also basically made of PET-G compounds. However, it has commercial grade additives for higher transparency, which are the main difference between the HDglass and PET-G filaments. The PLA filament has a semi-crystalline structure, resulting in smaller transparency than the other filaments. This filament was used for comparison purposes, since it is a commonly employed filament in 3D-printing processes. Moreover, the Tritan filament is a commercial compound made of copolyester aiming at a higher mechanical resistance and non-brittle behavior.

The first step in the POF fabrication is the 3D printing of its core, where the core is directly fabricated from the 3D printer nozzle. For the 3D printing of the optical fiber’s core, the Ender-3 V2 (Shenzhen Creality 3D Technology Co., Ltd., Shenzhen, China) 3D printer was used. In this case, the printing parameters were empirically determined in order to provide the same parameters for all filaments for comparison purposes of the different materials. It is important to mention that the parameters such as printing speed, nozzle temperature/cooling rate directly affect the pre-strain as well as the residual stress on the 3D printing of the polymer, especially the PET-G as depicted in [40]. However, the pre-strain analysis of the POFs is not in the scope of this paper, since the first step is to define a suitable filament for the 3D printing of the optical fiber. The parameters used on the 3D printing include a printing speed of 40 mm/s, a bed temperature of 80 °C and a nozzle temperature of 235 °C. Furthermore, the nozzle diameters of 1.75 mm and 0.4 mm were used to evaluate different diameters of the optical fiber core. For the fabrication of characterization samples, the layer thickness was set as 0.4 mm, and the fan speed set as 30%.

After the results of the refractive index characterization, the optical fiber core and cladding were fabricated. To be part of the core, the refraction index of the filament must be higher than the cladding. In addition, to present a better signal transmission, the values of transmittance must be close as possible to 100%. The filament that best fits these requirements is selected to make part of the core. The extrusion of the core is performed using the 3D printer nozzle. The filament and nozzle diameters used were 1.75 mm and 0.4 mm, respectively, and the extrusion temperature was set to 235 °C. Based on empirical testing, this temperature was chosen in order to minimize the production of bubbles during the fabrication of the core. The core diameter is controlled by using constant pulling force and printing speed. In addition, the fiber diameters are measured using a caliper every 10 mm of the 3D-printed POF.

To fabricate the cladding, we used two methods. The first is the dip-coating [5] using three different solutions: methyl acetate, sodium fluoride, and N,N Dimethylformamide. To study the cladding fabrication using this method, three immersion times were used for each solution: 6, 60 and 600 s. The purpose of this procedure is to make the ends of the core dissolve reducing its refractive index. The selected core was immersed at three different times, as shown in Table 1. The refractive indices were measured using a benchtop Abbe-type refractometer at around 585 nm.

The second technique to fabricate the cladding uses the resin Norland Optical Adhesive 88 (Norland Products Inc., East Windsor, NJ, USA) and a polytetrafluoroethylene (PTFE) tube with 0.6 mm inner and 0.8 mm outer diameters. After filling the tube with resin using a syringe, the core is inserted into the tube, immersing it. Then, the resin must be cured using ultraviolet light.

## 3. Experimental Setup

### 3.1. Optical Fiber Characterization

#### 3.1.1. Mechanical Characterization

The mechanical properties of the polymers were characterized by strain–stress testing, where it is possible to infer Young’s Modulus and maximum elongation. The tests are performed in tensile loading mode using the standard ISO 527-1:2012 [41]. In addition, the commercial 3D filaments used as the samples are positioned for the tensile stress applied using the Universal Testing Machine (Biopdi, Brazil), which has an input for an extensometer for the elongation assessment.

#### 3.1.2. Optical Characterization

For the optical characterization of the fabricated POF, three tests were conducted: refractive index characterization, transmittance characterization, and cutback test. For the refractive index characterization, a rectangular geometry of 15 mm × 25 mm with around 1 mm thickness was 3D-printed for each material. In addition, the samples were polished prior to the characterization to provide a suitable surface quality for the refractive index and transmittance/absorbance characterizations. The refractive index was measured in the visible wavelength range using a benchtop Abbe-type refractometer, where the samples are positioned inside the device and kept at a constant temperature throughout the measurement. Three measurements were performed with each sample. Then, for the transmittance characterization, the spectrophotometer Q898U2M5 (Quimis, Diadema, São Paulo) was used to measure the transmittance of the polymers with the same rectangular geometry. In this case, the samples are illuminated in the wavelength range of 350 nm to 850 nm, since there is the wavelength range with the smallest optical losses in polymer optical fiber materials. In addition, the transmittance at each wavelength is measured using a photodetector for each wavelength, where the sensitivity variation of the photodetectors as a function of the wavelength is normalized, and the results are presented from 0 to 100%, which is converted to absorbance and normalized as a function of the sample thickness.

The cutback method is applied for the optical losses assessment; this method involves measuring the optical power at different lengths of the POFs [42]. Using the relationship between the optical power and the fiber length, it is possible to estimate the attenuation. Thus, the fabricated 3D-printed POFs are used and, to perform this experimental characterization, a laser transmitter with a 3 mW optical power and 650 nm wavelength was used connected to one end of the fiber, whereas the other end is connected to the photodetector. Furthermore, the POF samples with 30 mm length are placed at a straight position, and the laser is focused on the fiber’s core, whereas the other end of the fiber is directly connected to the photodetector. Then, the samples are cut and the lengths are reduced to 10 mm and the transmitted optical power is measured again. The process is repeated once again for all samples.

### 3.2. Intensity Variation-Based Sensors

The 3D-printed POF is used in the development of a curvature sensor and a temperature sensor. For the curvature sensor development, the proposed sensor is obtained by fixing the fiber into a rotating joint, and measuring the optical power variation for each angle, as shown in Figure 1. The optical power was measured for a curvature angle variation ranging from 90 to 180° in steps of 15°. The optical power was measured using the phototransistor IF-D92 (Industrial Fiber Optics, Tempe, Arizona), as in [43].

Another approach is used in the development of a temperature sensor. The fabricated POF was immersed into a furnace used to raise the temperature from 30 to 40 °C in steps of 5 °C. For each temperature, the optical power was measured to characterize the temperature sensor. It is worth mentioning that such a small temperature range is related to the smaller repeatability and sensitivity of the temperature sensors based on the intensity variation principle without any geometry change for that purpose. Thus, the analysis is restricted to conventional room temperature variations.

### 3.3. Modal Interferometer

The 3D-printed POF is used in the development of a modal interferometer. In this case, the optical material with the highest transmittance (among the ones tested) is used as the sensitive region in the interferometer. The proposed interferometric sensor device is obtained from the splicing between the 3D-printed POF and standard single-mode fibers (SMFs). Thus, the modal interferometer works in the single-mode-multimode-single-mode (SMS) configuration as shown in Figure 2. The optical characterization is performed only on the visible wavelength region, since it is the region with the smallest losses for most of the POFs. However, for the interferometer approach, the 1550 nm wavelength region is used, since the interferometer is based on the single-mode-multimode-single-mode configuration in which there is a necessity to guarantee a single mode operation of the silica optical fiber coupled to the POFs, where such single mode operation only occurs in wavelengths higher than 1260 nm (cutoff wavelength of SMFs), whereas the multimode operation occurs on the 3D-printed POF. It is important to mention that the arrows presented in Figure 2 are only a schematic representation of the optical transmission in the interferometer. However, there are also reflections on the interface between the different fibers, which were not represented in the figure. Moreover, the interferometer configuration is related to the optical power transmission from SMF 1 to 3D-printed POF as well as the transmission from the 3D-printed POF to SMF 2, which leads to additional optical losses in the device. The optical losses due to the coupling between different fibers are around 3 dB, which leads to the necessity of using optical sources with higher optical power. Such losses are estimated from each coupling region. In this case, both optical fibers (SMF and 3D-printed POF) are placed in a 3D translation stage, where the SMF (with an FC/APC connector) is connected to a red laser and the optical power at the end facet of the POF is monitored as the fibers are aligned. The alignment process continues until the maximum optical power is obtained. Then, the UV-curing resin in the regions close to the fibers coupling region and the first stage of UV curing is performed using a UV lamp UTarget-365 (AMS Technologies, Germany) with 1400 mW/cm^2^ radiance at 365 nm for 2 s. Thereafter, the alignment between fibers is double checked by means of minor variations in the 3D positioning between fibers, after achieving the maximum optical power, the second stage of the UV curing is performed until there is a full curing of the resin. The losses due to the optical fiber coupling are estimated using the following steps. First, the optical power is estimated on the end of the SMF connected to the red laser. Then, after the UV curing of the coupling between the SMF and 3D-printed POF, the optical power is measured at the end of the 3D-printed POF. The measured optical power at the end of the POF is compensated by the attenuation of the optical fiber (already estimated after the fabrication of the 3D-printed POF). Thereafter, the optical power after the optical coupling process (compensated by the losses of the 3D-printed POF) is compared with the optical power measured before the optical coupling, which results in an estimation of the losses due to the coupling between SMF and 3D-printed POF. The process is repeated on the other end of the 3D-printed POF. It is important to mention that such optical losses at each coupling are similar to the ones reported in previous works that use a similar method for the optical fiber coupling [44]. The SMF used is the SMF-28 (Corning Inc., New York, NY, USA), whereas the multimode fiber is a 3D-printed POF, presented in this work, with an HD glass core (0.2 mm in diameter). To guarantee the single-mode operation of the SMFs, a broadband optical signal centered at 1550 nm is used (since it is above the cut off wavelength of the SMFs). However, POFs generally present high attenuation at such wavelength regions. For this reason, only 2-cm length of the 3D-POF is used, which also results in a small sensing region that can be beneficial for the analysis of concentrated parameters. It is important to mention that the coupling between the POF and SMFs is performed by the alignment between each component followed by the optical resin application to splice the POF and SMFs.

Considering the structure presented in Figure 2, the broadband optical signal transmitted through SMF-1 reaches the 3D-printed POF, a large multimode fiber in which there is a large number of propagating modes through the length of the 3D-printed POF. Such a large number of modes are combined into only one mode when the optical signal reaches SMF-2. For this reason, the interaction between the modes propagating at the 3D-printed POF with the temperature, strain or surrounding refractive index in the POF region leads to phase delays in some of the modes. Therefore, if the transmitted spectrum is evaluated after SMF-2, there is a wavelength shift in the interferometer signal as a function of the aforementioned variables.

The sensor is fabricated by splicing the 3D-printed POF in-between two SMFs. To that extent, the first step is to cleave each SMF to obtain a clear and plane end facet, i.e., without cleaving angles. Although it increases the Fresnel reflections, the plane end facet is preferred in this case, since the 3D-printed POF also has plane end facet (cleaved with a razor blade). Thus, it provides an accurate coupling of the fibers. After the cleaving processes, the fibers are aligned with the aid of a 3D-translation stage and a fault-detection red laser. After the alignment between the SMF-1 and 3D-POF, an optical adhesive NOA88 (Norland, USA) is used to attach both ends of the fibers. The optical adhesive is cured using the UV lamp UTarget-365 (AMS Technologies, Germany) with 1400 mW/cm${}^{2}$ radiance at 365 nm. Thereafter, the same process is applied to SMF-2 and the other end of the 3D-printed POF. However, the large core diameter of the 3D-printed POF leads to higher optical losses in the coupling between the POF and SMF2. The POFs are cleaved with a razor blade perpendicular to the end facet with sequential polishing using sandpaper with high grit size for a smooth surface. For this reason, a reduction of the transmitted optical power is mainly due to the second splicing, i.e., the POF splicing with SMF 2.

The proposed SMS interferometer is characterized as a function of the temperature and the strain using the setup presented in Figure 3, where a superluminescent light-emitting diode (SLED) DL-BP1-1501A (Ibsen, Denmark) with a center wavelength of 1550 nm, spectral width of 70 nm, and maximum optical power of 12 mW is used. The transmitted spectrum is acquired by an optical spectrometer I-MON 512 (Ibsen, Denmark) with 5 pm accuracy (https://ibsen.com/product/i-mon-512-usb accessed on 7 January 2023). For the temperature characterization, the sensing region is positioned inside a climatic chamber with temperatures varying from 25 °C to 45 °C with 5 °C steps, where the transmitted spectra are collected after a 5-min stabilization time. Considering the intensity variation also proposed, the interferometer approach generally results in higher temperature sensitivity and, for this reason, the higher temperature was applied for the characterization. Furthermore, the strain characterization is performed by gluing the sensitive region in a 1D translation stage comprised of a static and a movable stage attached to a micrometer. Sequential axial strains are applied to range from 0 μϵ to 250 μϵ in steps of 50 μϵ. All tests are performed three times for each condition.

## 4. Results and Discussion

### 4.1. POF Fabrication

After POF production, the diameters are measured along their length. Table 2 presents the average value and the standard deviation of POF diameters. The principal intent in this is to identify the regularity of the manufacturing process. The outer diameters of the fibers were measured using a caliper with 0.01 mm resolution at different regions of the fiber, where the mean and standard deviation of the measurements are presented.

The main aim of the immersion method is to reduce the refractive index from the outside to the inside of the core, producing the cladding, without reducing its diameter. For 0.4 and 0.2 mm filaments diameter, N,N dimethylformamide caused diameter reduction. This result indicates that N,N dimethylformamide is not suitable for cladding production using the immersion method, since the diameter reduction causes optical power loss and mechanical weakening. As the cladding fabrication failed, the data related to N,N dimethylformamide is not shown in Table 2. For optical adhesive, sodium fluoride, and methyl acetate, the fabricated POFs dimensions are shown in Table 2. The fiber diameters were measured using a caliper at every 10 mm of fiber. Then, the mean and standard deviation of the 3D-printed POFs are presented. The results show a small deviation in Sample 8, whereas the smallest diameter was found in Sample 4. It is important to mention that, even with such small dimensions, all fibers have a multimode operation due to the diameter difference and refractive indices of core and cladding. The samples with cladding made of optical adhesive presented the highest diameter due to the additional material applied on the fiber’s core.

### 4.2. POF Characterization

#### 4.2.1. Optical Characterization

The measured refractive index of the materials used in the fabrication of the POF can be found in Table 3.

The transmittance characterization of the used polymers is shown in Figure 4. Compared to the other analyzed polymers, HDglass had the highest transmittance in all the analyzed spectra from this characterization. As a result, this material is more suitable for fabricating the POF as it maximizes optical power transmission. As the goal is the assessment of the suitable material for POF fabrication, the transmittance assessment is performed with the filaments, where each sample was fabricated into a rectangular shape. Furthermore, the transmittance was measured using the fiber shape in all samples in a wavelength range from 350 nm to 850 nm due to smaller optical attenuation in POFs when such wavelength region is analyzed, which is generally used in POF applications. The samples have around 1 mm thickness. The results presented a dispersion in the transmittance considering all samples that can be related to deviations in the illumination of the sample. However, the deviations are similar to the ones obtained in previous works, such as [45].

The results obtained through the cutback method for the HDglass core without cladding are presented in Table 4, whereas the ones of HDglass with cladding fabricated with different approaches are presented in Table 5. The cutback method is performed by investigating the optical power variation at different lengths of the optical fiber. Thus, the first optical power (P1) is measured in length 1 (L1), whereas the second optical power (P2) is measured in length 2 (L2), presented in Table 4 and Table 5. In this case, a He-Cd laser with a central wavelength of 650 nm and 3 mW optical power and the photodiode IF-D92 (Industrial fiber Optics, Tempe, AZ, USA) are used as a light source and photodetector, respectively. For the light coupling between the tested fibers and the transmission/receiver components, a connectorless approach is employed. The fibers are placed at a straight position and the laser beam is focused on the optical fiber’s core, whereas the other end of the fiber is positioned in the housing of the photodetector for the optical power acquisition. Thus, the fibers are illuminated with a laser focused on the optical fiber end facet, whereas the other end is mechanically connected to the photodetector. Then, the fiber is cleaved using a razor blade perpendicular to the fiber end facet and the transmitted optical power is measured again.

As can be seen in Table 4, the 0.4 mm diameter core was able to transmit almost entirely the laser power in the tested lengths. On the other hand, the 0.2 mm diameter core has a much lower output power compared to the previous one, as well as showing that in a small interval of its length the attenuation indicates that there will be large losses. The difference between the attenuation observed in Table 4 for 0.2 and 0.4 mm core diameter is related to the connection between the laser, the optical fiber core, and the phototransistor. Accordingly, the 0.4 mm core diameter was more suitable for the setup, resulting in lower attenuation. However, it is important to mention that the optical fiber with 0.2 mm core diameter is more suitable for optical coupling with single-mode fibers. For this reason, the optical losses of the 3D-printed POF with the SMFs are lower when the 0.2 mm core is used, which lead to a trade-off between the optical fiber attenuation (lower for the 0.4 mm fiber) and the optical losses in the coupling between the POF and SMF. In addition, the results are presented with three decimal places in all cases (i.e., Table 4 and Table 5, which were obtained considering the optical power estimation).

Table 5 shows that the resin cladding improves the attenuation of the 0.2 mm core diameter fiber by one magnitude and, for tests performed with this diameter, the final beam power at the receiver did not exceed 50% of the reference value. As the goal of the paper was the preliminary analysis of the filaments prior to the POF development, there was no POF fabricated using the other materials mentioned, since they present higher losses indicated by the results in Figure 4. From the results obtained for samples 2, 6, 8, and 10, all transmitted the beam to the receiver maintaining the maximum power for the tested lengths and, as a consequence, the optical attenuation for the tested length was close to zero. Table 4 shows that only with the core was it possible to transmit the laser at a higher power than that presented in the results of Table 5 for sample 7. The same occurs when comparing the results of optical attenuation, which demonstrates that the immersed core in methyl acetate for 60 s was dissolved to the point of having a worsening in the transmission of the beam. The results of sample 9, at first, seem to be different from those found for immersions in the same solvent presented in samples 8 and 10. However, because the length of sample 9 is greater, it is possible to better evaluate the transmission loss characteristics of the immersion manufactured POF. The results presented in Table 5 show a reduction in the optical attenuation with the application of a cladding layer, where four samples (samples 2, 6, 8 and 10) show smaller attenuation than the ones presented in Table 4. Compared with previously presented results, the aforementioned Samples presented smaller losses than the ones presented in [45] (with optical losses around 1.2 dB/cm). The large differences in the optical losses between samples are related to minor variations in the 3D printing parameters as well as the deviations in the cladding fabrication, which can present deviations in thickness and composition. It is also worth mentioning that the 3D-printed POF samples generally present optical attenuation higher than the commercially available PMMA POFs (0.2 dB/m). However, some of the samples presented optical losses close to one of the commercially available POF with the additional advantage of higher customizability and a small time of fabrication.

#### 4.2.2. Mechanical Characterization

The Young modulus (*E*) and the maximum deformation ($\epsilon_{max}$) for each polymer, obtained by the mechanical characterization, are shown in Table 6. In order to determine the mechanical application for each material and to guide the selection of the best option for the application, these mechanical properties are considered in the selection of the suitable material for each application, i.e., materials with lower Young’s modulus have a higher sensitivity to mechanical sensors. It is important to mention that the elongation was measured in the region at which the stress–strain curves start to present a nonlinear behavior. In addition, the ultimate strength is also presented, where it is possible to observe a higher Ultimate strength and maximum deformation on the Tritan sample. In this case, the maximum elongation is more than two times higher than the one presented in HDglass. Moreover, the Tritan sample presented the smallest Young modulus, which indicates a potentially higher sensitivity in force/pressure assessment. Thus, the Tritan samples presented suitable mechanical properties for sensor applications. However, such properties must be combined with favorable optical properties for the development of POFs.

### 4.3. Curvature and Temperature Sensors Results

In the results of Figure 5 and Figure 6, it is possible to observe that there is linearity in the intensity variation. To verify the linearity, the determination coefficient ($R^{2}$) is used. This coefficient indicates how much one variable’s (curvature and temperature) variance explains the variance of the second variable (optical power). The characterizations show the high linearity of the sensors, which makes them suitable for sensors’ applications for the assessment of temperature and curvature.

As shown in Figure 5, the optical power reduces linearly ($R^{2}=0.969$) with the increase of the curvature angle. Since light beams tend to exit at the tangent of a curve, the greater the angle of curvature, the greater the loss in optical power, as observed in Figure 5. For the curvature angle sensor, the obtained sensitivity was 4.407 $\times \;10^{-4}$ mW/${}^{\circ }$. In addition, the temperature results presented in Figure 6, it is possible to observe the optical power variation as a function of the temperature. In this case, the optical power of the temperature sensor increases linearly with temperature ($R^{2}=0.956$). The sensitivity of the temperature sensor was 5.718 $\times \;10^{-3}$ mW/${}^{\circ }C$.

### 4.4. Interferometer Results

The transmitted spectrum of the proposed SMS interferometer is presented in Figure 7, where it is possible to observe the interference signal obtained in the 3D-printed POF region. The transmitted spectrum is normalized, since the spectrometer provides the signal in arbitrary units. It is possible to observe that there is attenuation on the transmitted signal due to the large attenuation of the 3D-printed POF at the 1550 nm region. In addition, the transmitted optical signal presented the periodicity commonly obtained in interferometers, where the differences as well as distortions in the spectrum are related to the modal interactions of 3D-printed POF due to its large multimode operation. Nevertheless, the proposed interferometer can be readily employed in sensing applications, since the spectrum can be acquired with commercial spectrometers with little to no pre-processing of the optical signal. It is important to mention that only the HDglass was used on the interferometer development, since such core material presented the smallest losses in the 3D-printed POFs among the ones tested. Furthermore, the samples presented in Figure 7 are the ones with the smallest optical losses. However, the additional losses in the coupling between the 3D-printed POF samples and the SMFs lead to an additional reduction in the transmitted optical power, resulting in small amplitude variations.

Regarding temperature characterization, Figure 8a shows the transmitted spectra of the proposed sensor at different temperature conditions. The results indicate a wavelength shift of the transmitted spectrum in which all peaks presented similar sensitivities, indicating that the spectrum moves in unison. For this reason, linear regression is performed considering only the peak at 1548 nm. Figure 8b shows the linear regression using one of the peaks obtained in the transmitted spectrum of the proposed SMS interferometer, where it is possible to observe a blue shift of the spectrum as a function of the temperature. Such behavior is different from the ones commonly obtained using only silica-based optical fibers in which there is a red shift as a function of the temperature [46]. The reason for the such a blue shift in the spectrum is the negative thermo-optic coefficient commonly obtained in acrylate-based POFs [46]. Nevertheless, the temperature sensitivity of 78.8 pm/°C indicates a sensitivity some orders of magnitude higher than the ones commonly obtained in fiber Bragg gratings [47], which also result in higher resolution for temperature detection.

Similarly, the transmitted spectra of the SMS interferometer under strain are also obtained as shown in Figure 9a, where it is possible to analyze the wavelength shift as a function of the applied strain. Once again, all peaks in the spectrum presented similar sensitivities, which indicates that the spectrum moves in unison as a function of the axial strain as well. The linear regression indicated in Figure 9b shows a determination coefficient ($R^{2}$) of 0.962, which can be enhanced if a polynomial regression (order 2) is applied, resulting in an $R^{2}$ of 0.991. The strain sensitivity of the proposed sensor is analyzed as a function of linear regression (since the sensitivity is not constant in polynomial regressions). Such sensitivity analysis indicates a sensitivity of 9.5 pm/μϵ, which is also orders of magnitude higher than other optical fiber sensor approaches. Moreover, the error in the sensor response is below 5%.

The temperature and strain results indicate the feasibility of the proposed approach on the straightforward development of highly sensitive sensors using low-cost approaches, such as 3D-printing and commercially available fibers and components. The fabrication is simple, highly customizable, and results in sensors that present high resolution.

## 5. Conclusions

This work presented an alternative production method for POFs using simplified methods and low-cost materials. The POF fabrication is performed through 3D printing using four different commercially available filaments. As a result of the printer nozzle, the core fabrication showed a high degree of regularity in their diameters (maximum standard deviation is 0.03 mm, representing 3.6% of the average diameter of the produced sensors). It is also possible to manufacture POFs with various mechanical and optical characteristics using AM methods, primarily by studying the raw polymer used to produce the POFs. This work uses a cutback method that is representative of short-range applications, such as sensor applications and data transmission over short distances. To understand the limitations of POFs with regard to attenuation for long-range applications, this characterization should be repeated in further studies with POF longer lengths. The proposed 3D-printed POF can be used in the intensity variation-based principle for the measurement of curvature angle or temperature due to a linear optical power variation as a function of these parameters. In curvature angle assessment as a function of the optical power, $R^{2}$ of 0.969, whereas $R^{2}$ of 0.956 was obtained for the linear regression between transmitted optical power and temperature. Further research could be conducted to utilize this POF fabrication method to produce POFs for other sensing applications, such as refractive index sensors and pH sensors. The fabricated POF could also be used as an interferometer to measure temperature with an $R^{2}=0.979$ with a sensitivity of 78.8 pm/${}^{\circ }C$ and strain with an $R^{2}=0.962$ with a sensitivity of 9.5 pm/μϵ. In addition, future works include the investigation of printing parameters on PET-G filaments for the assessment of the correlation between the printing parameters and the pre-strain and residual stress in fiber fabrication.

## Figures and Tables

**Figure 1 polymers-15-00640-f001:**
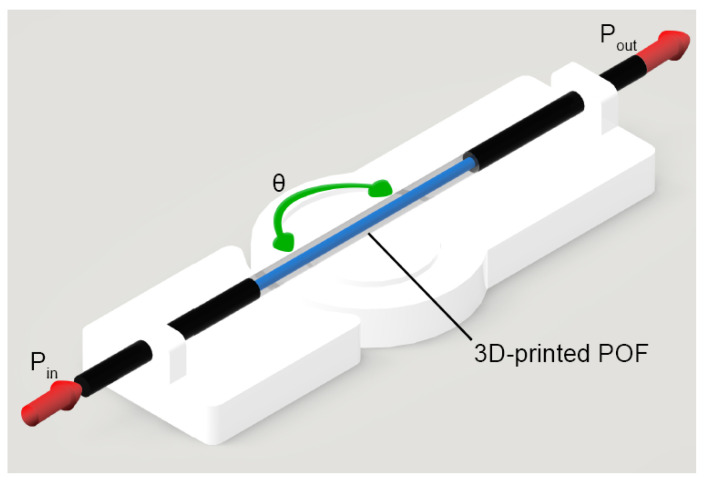
The setup used to measure the change in the intensity of the laser beam with the variation of the angle θ.

**Figure 2 polymers-15-00640-f002:**
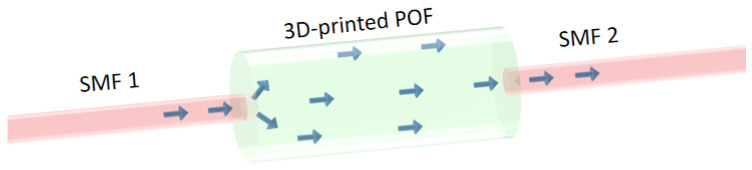
Fibers’ mode propagation difference.

**Figure 3 polymers-15-00640-f003:**
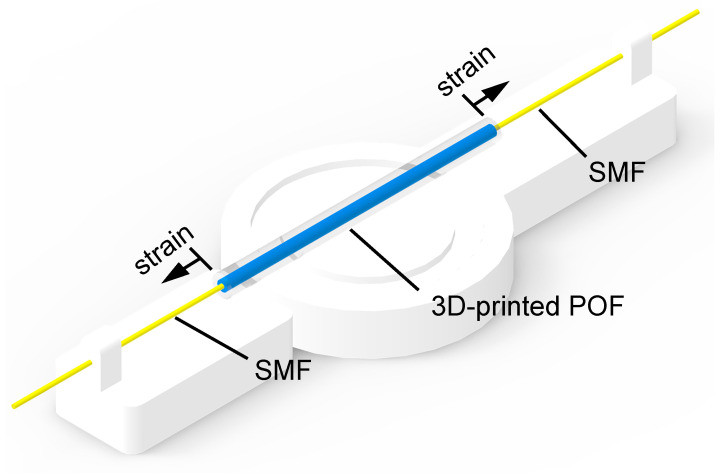
Interferometer strain measure setup.

**Figure 4 polymers-15-00640-f004:**
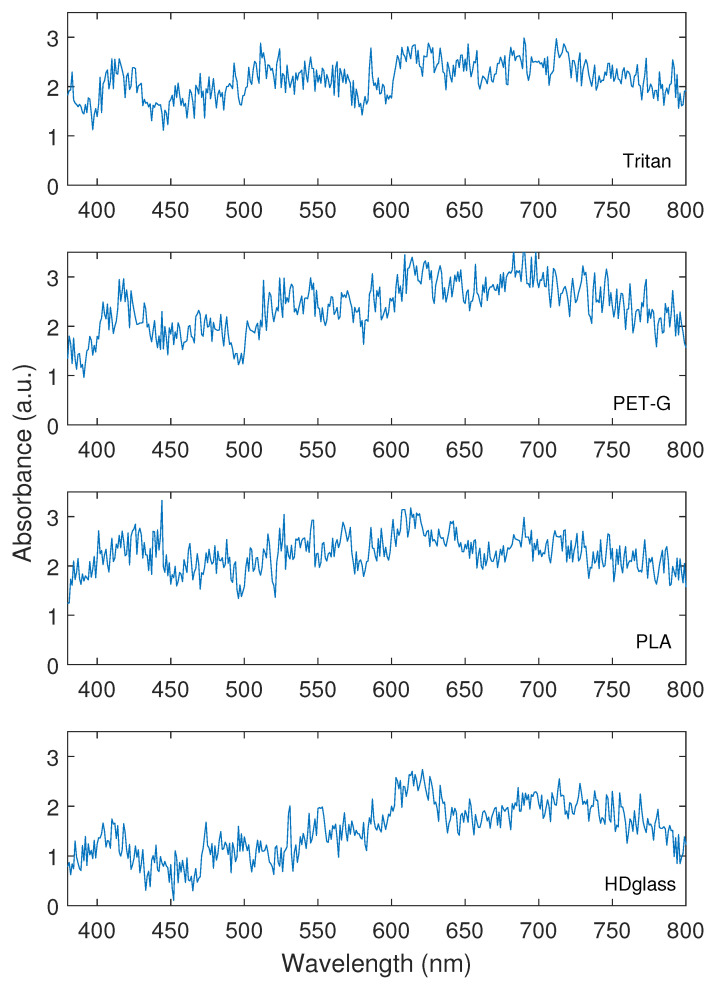
Filament absorbance values measured in the spectrophotometer for the wavelengths where the POFs have the smallest attenuation.

**Figure 5 polymers-15-00640-f005:**
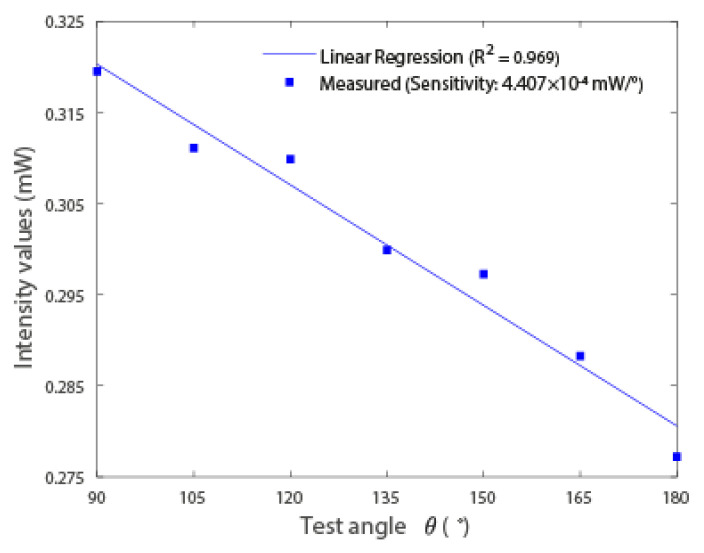
Intensity results (normalized between the minimum and maximum intensity values acquired in each test) in relation to the variation of the angle between ends.

**Figure 6 polymers-15-00640-f006:**
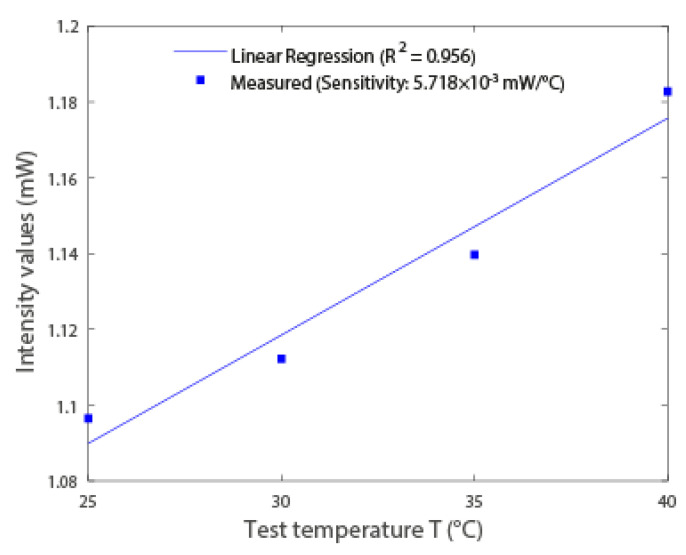
Intensity results (normalized between the minimum and maximum intensity values acquired in each test) in relation to temperature variation.

**Figure 7 polymers-15-00640-f007:**
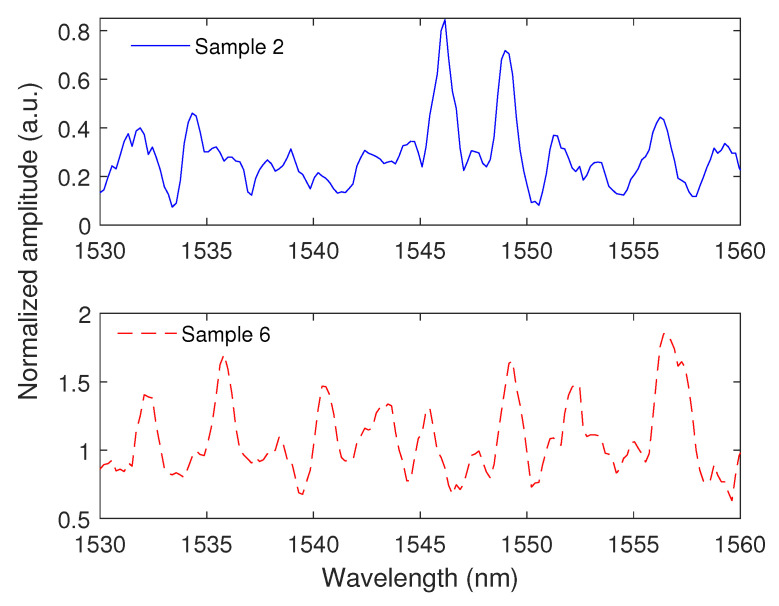
Transmitted spectrum (with normalized optical power) of the proposed SMS interferometer using two samples of the 3D-printed POF.

**Figure 8 polymers-15-00640-f008:**
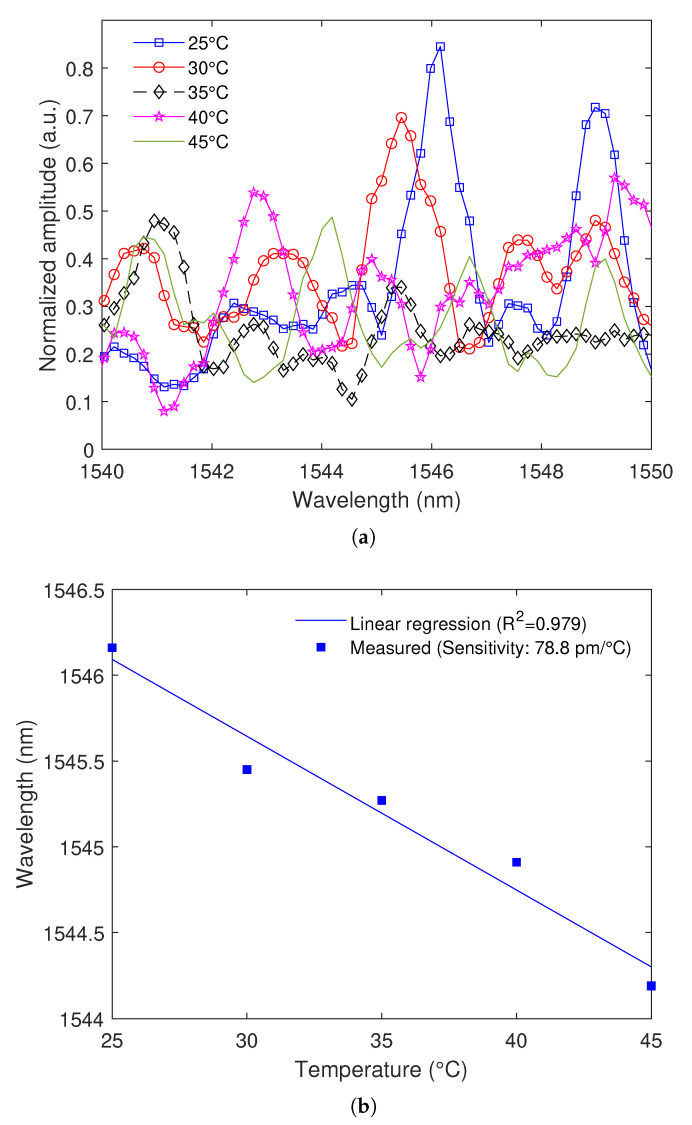
(**a**) Transmitted spectra as a function of the temperature; (**b**) Wavelength variation as a function of temperature.

**Figure 9 polymers-15-00640-f009:**
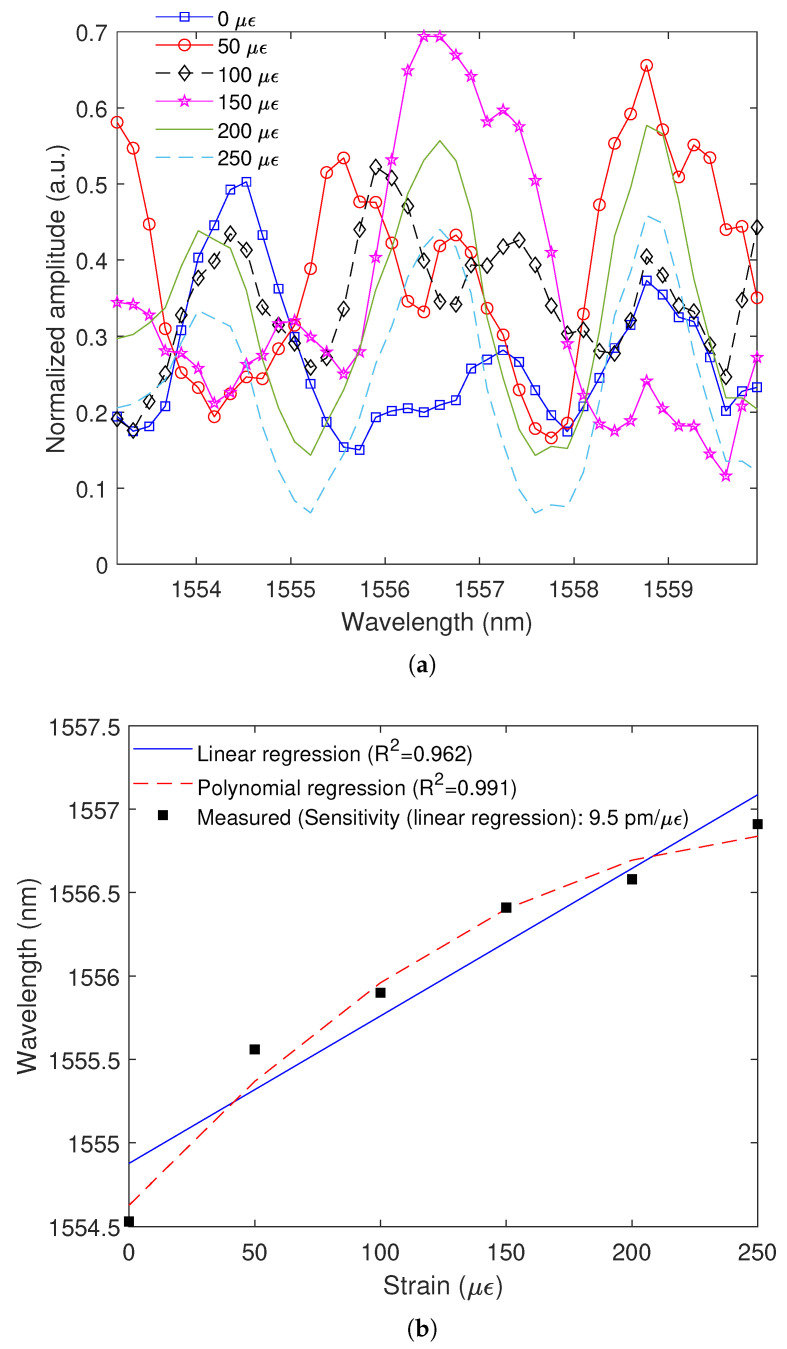
(**a**) Transmitted spectra as a function of the strain; (**b**) Wavelength variation as a function of strain for linear and polynomial regressions.

**Table 1 polymers-15-00640-t001:** Time of immersion of all solutions.

Sample	Solution	Time (s)	Refractive Index
1	Methyl acetate	6	1.361
2	Methyl acetate	60	1.361
3	Methyl acetate	600	1.361
4	Sodium fluoride	6	1.325
5	Sodium fluoride	60	1.325
6	Sodium fluoride	600	1.325
7	Dimethylformamide	6	1.4305
8	Dimethylformamide	60	1.4305
9	Dimethylformamide	600	1.4305

**Table 2 polymers-15-00640-t002:** Produced samples after cladding fabrication.

Sample	Core Diameter (mm)	Cladding	Average Diameter (mm)	Standard Deviation (mm)
1	0.2	Optical Adhesive	0.83	0.02
2	0.4	Optical Adhesive	0.84	0.03
3	0.2	Sodium fluoride 6 s	0.20	0.01
4	0.2	Sodium fluoride 60 s	0.19	0.01
5	0.2	Sodium fluoride 600 s	0.20	0.02
6	0.4	Methyl acetate 6 s	0.42	0.02
7	0.4	Methyl acetate 60 s	0.39	0.03
8	0.4	Sodium fluoride 6 s	0.44	0.00
9	0.4	Sodium fluoride 60 s	0.45	0.01
10	0.4	Sodium fluoride 600 s	0.44	0.02

**Table 3 polymers-15-00640-t003:** Measured refractive index values at around 585 nm.

Material	Refractive Index
HDglass	1.645
PET-G	1.655
PLA	1.648
Tritan	1.646
Optical Adhesive	1.56

**Table 4 polymers-15-00640-t004:** Attenuation calculated to HDglass cores without cladding. The attenuation was measured at 650 nm wavelength region.

Diameter (mm)	L1 (mm)	P1 (mW)	L2 (mm)	P2 (mW)	Attenuation (dB/cm)
0.2	69.60	0.05	60.67	1.11	3.62
0.4	66.00	2.97	56.86	3.00	0.05

**Table 5 polymers-15-00640-t005:** Attenuation calculated to produced POFs. The attenuation was measured at 650 nm wavelength region.

Sample	L1 (mm)	P1 (mW)	L2 (mm)	P2 (mW)	Attenuation (dB/cm)
1	30.68	0.52	25.13	0.58	1.23
2	49.00	2.99	39.67	2.98	0.02
6	40.00	3.00	35.11	2.99	0.03
7	54.30	2.12	49.11	3.00	2.94
8	55.00	3.00	48.21	2.99	0.03
9	63.50	0.45	57.20	2.65	12.33
10	47.40	3.00	41.22	2.99	0.10

**Table 6 polymers-15-00640-t006:** Mechanical properties calculated from data acquired in stress–strain tests.

Property	HDglass	Tritan	PET-G	PLA
*E* (GPa)	1.6	1.3	1.7	2.4
$\epsilon_{max}$ (%)	1.2	2.5	1.0	0.8
Ultimate strength (MPa)	50	53	33	31

## Data Availability

Not applicable.

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
