# Peer review of "Three-Dimensional-Printed Fabrication of POFs Using Different Filaments and Their Characterization for Sensing Applications"

_polymers, 2023, doi:10.3390/polym15030640_

Round 1

Reviewer 1 Report (Previous Reviewer 1)

All referee comments are well addressed. The only suggestion that can increase the quality of the article is to add the potential applications of these materials, which you can use from the following sources.

https://doi.org/10.1002/masy.201900074

https://doi.org/10.1177/15589250211052766

Polymer optical fiber sensors—a review

Author Response

Dear Reviewer 1,

Please, find attached the answers to your report.

Thank you very much for your comments.

Kind regards,

Reviewer 2 Report (Previous Reviewer 2)

The manuscript is in much better state than before. Just few comments;

The authors should add more detail on the novelty of the work and describe the aims in detail prior the start of experimental section. The conclusion part is should be exclusive to the main findings. 

Author Response

Dear Reviewer 2,

Please, find attached the answers to your report.

Thank you very much for your comments.

Kind regards,

Reviewer 3 Report (Previous Reviewer 3)

For comments see attached file.

Author Response

Dear Reviewer 3,

Please, find attached the answers to your report.

Thank you very much for your comments.

Kind regards,

Round 2

Reviewer 2 Report (Previous Reviewer 2)

no comments

Author Response

Dear Reviewer 2,

Please find attached our modifications based on the reviewers’ comments of the manuscript entitled “3D-printed Fabrication of POFs using Different Filaments and their Characterization for Sensing Applications”. 

Each of the authors has read and agreed with the current version of the manuscript. This manuscript has not been published or submitted for publication to any other journal.

Best regards,

Robertson Pires-Junior
Leandro Macedo
Anselmo Frizera
Maria José Pontes
Arnaldo Leal-Junior

Federal University of Espirito Santo, Brazil

Reviewer 3 Report (Previous Reviewer 3)

For comments see attached file.

Author Response

Dear Reviewer 3,

Please find attached our modifications based on the reviewers’ comments of the manuscript entitled “3D-printed Fabrication of POFs using Different Filaments and their Characterization for Sensing Applications”. 

Each of the authors has read and agreed with the current version of the manuscript. This manuscript has not been published or submitted for publication to any other journal.

Best regards,

Robertson Pires-Junior
Leandro Macedo
Anselmo Frizera
Maria José Pontes
Arnaldo Leal-Junior

Federal University of Espirito Santo, Brazil

Round 3

Reviewer 3 Report (Previous Reviewer 3)

for comments see attached file

Author Response

Dear Reviewer 3,

Please find attached our modifications based on  your comments of the manuscript entitled “3D-printed Fabrication of POFs using Different Filaments and their Characterization for Sensing Applications”. 

Best regards,

Robertson Pires-Junior
Leandro Macedo
Anselmo Frizera
Maria José Pontes
Arnaldo Leal-Junior

This manuscript is a resubmission of an earlier submission. The following is a list of the peer review reports and author responses from that submission.

Round 1

Reviewer 1 Report

This article uses four different materials as polymer optical fibers (POFs). Two coating methods have been used for cladding, and optical and mechanical tests have been presented. There are fundamental uncertainties in the novelty, selection of materials, the use of 3D printing or filament printers, the tests performed, and the analysis of the results.

1.      In relation to the research material, it should be clarified in the abstract, title, and conclusion sections. In the title of the article, only PETG is used. While in the section on materials, research methods, and results, four commercial materials are presented.

2.      In this case, these materials should be provided in more detail. What is the reason for choosing these materials?

3.      I do not understand what Tritan is. And what is the difference between PETG and HD Glass (a commercial PETG filament from FormFutura)?

4.      Which company made the filament used? It is mandatory to mention the exact specifications of the raw material. At least, if the physical, chemical or mechanical characteristics of the filament are not provided, the name of the manufacturing company and its grade should be mentioned.

5.      PLA cannot have high transparency due to its semi-crystalline structure. While the authors mentioned that four transparent materials were used.

6.      The novelty of the research should be clearly stated. For this purpose, the following sources should be considered.

https://doi.org/10.1016/j.yofte.2020.102299

https://doi.org/10.1016/j.addma.2022.102996

https://doi.org/10.1109/LPT.2022.3175803

7.      The images used in Figure 1 do not match, and it is suggested that this figure be deleted. Part a can be provided if it is accompanied by scale.

8.      In the mechanical characterization section (3.1.1 page 3), there are fundamental issues that should be corrected. In which loading mode was the mechanical test done? What is the standard used? What is the loading rate? How to measure elongation? Has a precision instrument such as Digital image correlation or extensometer been used?

9.      In the optical characterization section (3.1.2 page 4), more details regarding the performed tests should be provided. The geometry of each optical test should be presented.

10.  Why are the printing parameters not presented? It is suggested that all printing parameters, such as nozzle temperature, speed, layer thickness, nozzle diameter, and bed temperature, be presented. How are the printing parameters selected?

11.  I don't understand the connection between the article and 3D printing. In which section manuscript 3D printer or additive manufacturing is used?

12.  Printing parameters strongly affect mechanical, optical, and physical properties. Especially the parameters that affect the amount of pre-strain during the printing process and the cooling rate, such as the speed and nozzle temperature. Are these effects considered? Or, according to the results, can you predict their effect? For more details, use the following paper.

https://doi.org/10.1016/j.jmrt.2022.03.121

13.  The 3D printing of PETG with the FDM method has received a lot of attention in recent years and is one of the practical materials in this field. The following sources are suggested for use in different parts of the article.

https://doi.org/10.1088/1361-665X/ac77cb

https://doi.org/10.3390/app10093062

https://doi.org/10.1016/j.mfglet.2022.05.002

https://doi.org/10.3390/polym13111758

https://doi.org/10.1016/j.jmrt.2022.04.076

https://doi.org/10.3390/polym14132564

14.  The presented mechanical properties results (Table 6, page 9) are ambiguous. At what point elongation values are extracted (yield point, ultimate strength, or failure)? Why yield and ultimate strength values are not provided?

15.  In the results and discussion section, there is no analysis and discussion. only the obtained data are presented and reported. If so, each of the results should be analyzed and also compared with results of related papers.

Reviewer 2 Report

It is an interesting work, however, requires revisions.

1.      The overall writing should be improved.

2.      Introduction part is too small in terms of smart textiles, it should be expanded with appropriate reviewed papers, followings can be added;

a.      A fiber Bragg grating-based smart wearable belt for monitoring knee joint postures

b.     Investigating flexible textile-based coils for wireless charging wearable electronics.

3.      How the diameter size was controlled?

4.      Have authors think about including other parameters?

Reviewer 3 Report

for comments see attached file
